# Language Should Reflect Biological Knowledge

**DOI:** 10.3390/ani15172476

**Published:** 2025-08-23

**Authors:** Donald M. Broom

**Affiliations:** St Catharine’s College and Department of Veterinary Medicine, University of Cambridge, Madingley Road, Cambridge CB3 0ES, UK; dmb16@cam.ac.uk

**Keywords:** language misuse, anthropocentrism, animal, innate, heart, welfare, sustainability

## Abstract

In order for the functioning of humans, of other animals and of other organisms to be understood better in the future, there is a need for language to change so that biological errors are not made. Words like animal, instinct, heart, pest, bacteria, health and sustainability are among those discussed in this paper because they are often misused. It should be possible to change language in order for human and other life to be able to be improved. A better understanding of biology is important for future generations.

## 1. Introduction

Language does not necessarily reflect society [1] but the discoveries that the earth is not flat and is not the centre of the universe did eventually change human language and thinking. Problems arise when changes in knowledge are not followed by changes in language. In some areas of knowledge there has been much discussion about thought, language and society but, in relation to science and to biology in particular, inaccurate usage of words is widespread and the misuse is often ignored by the general public and by some academics but has been challenged by others. For example, Derrida [2,3] criticises those who suggest that there is ‘opposition’, meaning a major difference between humans and other animal species, and suggests use of the word ‘animot’. Although still human-centred in his approach, he refers to how non-human animals might view humans and argues that a better understanding of humans comes from such thinking. The principal aim of this paper is to point out the extent to which words with a precise biological meaning are misused and to consider some of the negative consequences of such misuse for the world.

The results of many aspects of scientific research should help the way in which people think and change the words that they use. Some scientific discoveries and developments are rapidly incorporated in language but others may conflict with long-held views and be deliberately suppressed or just ignored. In particular, language has not kept up with biological knowledge. The failure to change thinking and language is not a trivial matter as it can greatly alter human practices, including the actions of governments. Language usage has affected attitudes to women and to non-white people. Some sexist and racist words have perpetuated prejudice and some have led to changes in laws. The impact of the use of speciesist words may also have been great [4]. Some of the biological terms that are in frequent use as part of the vocabulary of scientists and non-scientists and whose misuse impedes understanding, especially when this causes harms, are discussed here. Strickland [5] describes “cross-linguistic regularities” and presents the interesting concept of “core knowledge” which requires biological analysis beyond the scope of this paper. When language is used during communication, a word with a particular meaning can sometimes be used in an entirely different way or as a sub-category of its full meaning. The word may be defined in law, refer to human food or another resource, obscure something that the hearer might find unpleasant, or just be a pejorative term. Some words are misunderstood and misunderstandings may be perpetuated over human generations, during which time scientific knowledge changes but language may not change. The examples given below refer to English words but the concepts, prejudices and errors are relevant to all languages.

## 2. “Animal” and Terms Used for Animals

The way in which the word “animal” is used affects attitudes and actions [6]. Humans are animals. This simple sentence is understood in part by most people but the word animal is often erroneously used to distinguish between humans and other animal species. Logically, “humans and animals” should always be “humans and other animals”. An animal is a living being with a nervous system and other complex mechanisms for obtaining energy, using energy, and reproducing. Humans are a sub-category of animals [7] and the use of the unqualified word animal should always include humans. There is only one biology and almost every aspect of biological processes in humans is identical in some other animal species [7,8]. The similarities between humans and other animal species are great and the differences are small but there is a widespread view that humans are “special” and enormously different. Even in brain function, humans are similar to other sentient animals but to refer to a person as an animal sometimes implies being aggressive or uncontrolled as if these are not human characteristics but are typical of non-human species [7].

A second mistake in the use of “animal” is to imply that it refers only to mammals. People may say “animals and birds” implying that birds are not animals. Similarly, fish, molluscs and insects are often spoken of as if they were not animals. Indeed, those animals that are unlike mammals and birds, in that they are cold-blooded, are sometimes thought of as having very inferior functions, for example, no pain or other emotions and feelings. Biological research shows that much sophisticated functioning is common to cold-blooded and warm-blooded animals and many of these species are sentient [9]. These usages of the word animal have the potential to diminish the value attributed to individual animals as compared with the values of humans or mammals.

In the laws of many countries, all non-human animals are categorised as things in contrast to persons. By this action they are contrasted with persons but they are also grouped with objects that are not living. Most persons, on the other hand, differentiate between animals and things and distinguish living organisms from inanimate objects. Laws that refer to animals only in relation to human usage also pose problems in that they may suggest that the biological functioning and potential for poor welfare of an individual of a species is very different according to its usage when this is not biologically sound. For example, a pet rabbit, a laboratory rabbit and a wild rabbit will suffer in a similar way when subjected to a particular injury [10]. An idea gaining ground is that the animals used by people should be in a different legal category from things and that sentient animals, in particular, should be treated in a way nearer to how humans are treated. Such a change in category is more consistent with biology than would be a change in the status of some non-human animals to be included as persons. In scientific literature, the word person is almost always restricted to *Homo sapiens* so a chimpanzee or an elephant cannot logically be considered a person. Changes in law have usually been slow to occur but a rapid change is now needed.

The words used to refer to those animals used by humans for food and other resources sometimes have the objective of making it easier for the human user to have negative effects on the welfare of those animals by reducing emphasis on life qualities, species characteristics or individuality. When animals are always spoken of collectively and never as individuals, each individual is devalued. Terms, such as ‘veal calf’, that refer to a food product when identifying a living individual, or calling the individual “it” rather than “he” or “she”, also have this effect. The wording of references to experimental animals in scientific journals has often been aimed at emphasising the scientific or commercial value of that use and minimising evidence of the cost to those individual animals [11]. For farmed or wild animals used for human food, the word harvest, which is very much a term relating to plant production, is sometimes used and this makes the human action seem to be directed towards organisms that would not be subject to poor welfare if harmed. A word that is used by humans about certain other humans is “slave”. There are parallels between much human usage of other animal species and slavery [12]. There is also debate about whether or not it is accurate for those people who live with a companion animal to describe that animal as a friend or family member.

When animals used by people are killed, the words used to refer to this can sometimes be inappropriate. Birke et al. [13] pointed out that the use of the word “sacrifice” to describe laboratory animals killed for experimental purposes has impacts on the user and on the reader of the words. Words should be used in the same way for humans and for non-humans, for example “euthanasia”, meaning originally a good death, is used in relation to humans solely to mean killing an individual for the benefit of that individual and in a humane way. If the benefit is for someone else, as it may be when a dog is no longer wanted or if a laboratory animal is not wanted because an experiment has been concluded, the action should be called killing or humane killing but not euthanasia [14,15].

The use of the names of animals as terms of abuse for people or their actions, for example, pig, bitch, cow, donkey, snake, shark, ratted on him, is biologically incorrect and harmful to human attitudes to those animals [7]. Should companion or other animals be given individual names by people? There is an advantage of doing so in that the use of an individual name for any individual, human or non-human may increase respect for that individual [16]. However, referring to an individual by a name that it does not recognise can diminish that individual in the estimation of other people so there is an argument for limiting the use of individual names to circumstances where the individual concerned has the capability and opportunity to recognise and respond to that name.

## 3. Genetically Determined, Instinctive, Innate

The ethology and psychology literature has long indicated that there are very many ways in which the environment of an individual affects behaviour, and indeed every characteristic of animals [17]. Both nurture and nature have effects. Although all characteristics of plants and animals are affected by genetics, recent evidence, including that from epigenetics research, shows that every stage in the expression of genetic material, even the initial production of mRNA, can be affected by environmental factors [18,19,20]. Hence nothing is entirely determined by genetics, nothing is “instinctive” and nothing is “innate” where these terms mean independent of environment [21,22]. As a consequence, these words should not be used and it is incorrect for a person to excuse their own anti-social behaviour as being unavoidable due to their genes. Neither is it correct to say that any breed of dog is always aggressive. The concepts of genetic determinism, genetic essentialism and genetic reductionism, together with some of their negative consequences, are discussed by Hardern [23].

## 4. Heart, Other Organs, Bodily Integrity

It was long thought that the “heart” is the part of the body where knowledge and emotions are located but it has been known for many years that the brain is that location. However, people still say “I know in my heart” or “my heart tells me”. Other expressions refer to feelings emanating from other parts of the body, for example: “I feel it in my gut”, “I feel it in my bones”, or “it is in his blood”. There are some heart–brain–gut interactions. Congenital heart disease can have consequences for gut microbiome and function [24]. Feelings can change heart function and other physiological processes and awareness of heart functioning can change feelings. Individuals can use changes in behaviour and in physiological functioning, such as [25] slowing or accelerating heart-rate, to modify motivational state and hence subsequent behaviour and ability to cope with their environment [26,27]. Whilst it may be argued that there is a need for a term to use when referring to feelings, it would be better if this were not biologically erroneous. Rather than saying that feeling is in the heart, gut or bones it would be better to use “emotional brain” or to coin a new term.

The concept that preservation of the “integrity of the body” is important but has been understood by some people to imply that any addition to or subtraction from the body is fundamentally objectionable. As a consequence, injection of anything into the body has sometimes been perceived as an intolerable violation. This view may promote distrust of some valuable medical actions. The idea that vaccination is unacceptable because it involves introduction of foreign material into the body has a long history [28]. However, the scientific evidence for the very great positive consequences of vaccination for the vaccinated individual, for the family and friends of that individual and for the general community, is clear. Statements that vaccination is bad for you have caused great harm, as is evident from the many children who have suffered or died from measles because their parents did not allow them to be vaccinated and the many thousands of people who died from COVID-19 because of those who refused to be vaccinated. Some of those who died were infected by people who refused to be vaccinated and misunderstanding of biological words and concepts had tragic consequences for the non-vaccinated and for others in society.

## 5. Cognitive Ability of Animals

People who use non-human animals in ways that will, or might, have negative effects on those animals sometimes choose language that helps them to deal with such a situation by belittling the animals. This may be done by using words that imply that they have little or no ability or complexity of functioning. Even those people who derive no benefit from use of such animals may repeat statements that suggest that the gap between humans and all other species is very great, for example, exaggerating or assuming differences in cognitive ability. An example of this is to say that sheep are “stupid, silly or dumb.” The sheep observed may be attempting to escape from the humans and the person using the words may or may not know that sheep have complex social behaviour and that in learning tasks sheep can do better than dogs [9,29]. However, when members of the public in the United States were asked about wild mammals, a substantial proportion held the view that their cognitive ability was high although fewer ascribed the capacity for emotions to them [30]. Many members of the public are interested in evidence for non-human animals being clever [31,32] but show reluctance to change language accordingly.

## 6. Negativity About Small Animals

Many people think of all flies, most other insects, spiders and some rodent species in a negative way and may deliberately harm them. This is partly because the animals are small and thought of as simpler and of less importance in the world than human-sized animals. The words used to describe them may diminish them or misrepresent them. They are often referred to as pests, a term that groups animals that are thought to transmit human disease or to have negative effects on crop production or other human activities. In reality, only a very small proportion of species of small animals have any negative effect on people. A lack of understanding of flies results in some people saying that: “all flies carry diseases”, “flies come into the house to annoy me and to eat or contaminate my food”. In fact, very few species of fly carry any human disease and most flies enter houses by accident and are likely to die of dehydration unless helped to escape. These attitudes to flies [33] and to other small animals [34] lead to immoral actions such as killing them. Negative attitudes to insects make it harder to persuade the public that action should be taken to change agricultural and other practices in order to counteract insect biodiversity loss [35].

## 7. Bacteria and the Concept of Germs

The idea that small is malevolent and probably dangerous is even more pronounced in attitudes to bacteria. There is a general assumption by the public that all bacteria are pathogenic and harmful to humans. Most people are unaware of the positive role of bacteria in most ecosystems and are surprised to learn that there are more bacterial cells than human cells in the human body [36]. These attitudes are perpetuated in part by the word “germs”. The extremely human-centric view that most people have impairs their understanding of the diversity of bacteria even more than it affects views of the diversity and value of multicellular animals.

Anti-microbial resistance is an enormous threat to humans. Strains of bacteria that are resistant to antibiotics, such as some strains of the *Mycobacterium* that causes tuberculosis in humans, could cause the deaths of a substantial proportion of the world’s population in the near future [37]. Much of this situation has arisen because of failure to understand the term “anti-microbial resistance” and the misuse of medicines by much of the general public, for example, patients demanding antibiotics to treat viral and other non-bacterial diseases, patients failing to complete courses of antibiotic, people disposing of antibiotics into sewage, and people using antibiotics for growth promotion or unnecessary prophylaxis in farm animals.

## 8. Welfare, Health and Pain

Scientific writings during the last 40 years about the welfare of individual animals, human or not, have emphasised that welfare can be measured using a range of methods and that welfare varies from very positive to very negative. The concept of welfare includes all attempts to cope with the environment, not just feelings or just health, or just those aspects that promote productivity in farm animals [7,38,39,40]. The welfare of an individual is its state as regards its attempts to cope with its environment [22,41]. A nervous system is required to do this so the word “welfare” is used only for animals. The word welfare is still often used by the public as if it referred only to the positive but the balancing of positive and negative aspects [42] clearly occurs. Welfare is readily related to other concepts such as stress, health, awareness, consciousness and sentience [22,43]. Health is clearly an important part of welfare, not something separate from it, so it is not logical to say “health and welfare”.

The terms ‘natural’ and ‘naturalness’ which have multiple meanings and can have different connotations for different people, have been used in ambiguous ways in relation to welfare [7]. Rollin [44,45] advocated that ‘animals should be able to lead reasonably natural lives’ and both Rollin and Fraser [39,46,47] stated the importance of understanding animal needs. These authors did not say that naturalness contributes to a definition of welfare or should be part of welfare assessment. The state of an individual trying to cope with its environment will necessarily depend upon its biological functioning or, put another way, upon its nature. While natural conditions have affected the needs of the animal and the evolution of coping mechanisms in the species and the environment provided should fulfil the needs of the animal, that environment does not have to be the same as the environment in the wild [48]. Conditions in the wild may result in starvation, disease and predation, with consequent very poor welfare [49]. When the view is expressed that what is natural should be considered in relation to welfare, this is correct, but the concept and definition of welfare do not include naturalness [7,50].

Health, like welfare, can be qualified as good or poor and varies over a range. It refers to body systems, including those in the brain, that combat pathogens, tissue damage or physiological disorder so health can be defined as the state of an individual as regards its attempts to cope with pathology [51]. Health can be negative or positive so the health of every human or other animal can be considered and it is not logical to say “there is no health in us”. The scientific, precise meaning of the word health is better to use than archaic wording, including the description by WHO in 1946, written before the current meaning of welfare was used. Health refers to an individual coping with pathology so it is only applicable to living organisms and should not be applied to inanimate objects, to the environment or to the planet.

Pain is an aversive sensation and feeling associated with actual or potential tissue damage [52]. However, many people use “pain” to mean any negative experience. This usage is sufficiently imprecise to be misleading as well as erroneous. A further biological misunderstanding that causes confusion is the idea that pain is very different in humans and other vertebrates or invertebrates. The way in which the pain system functions is similar across a wide range of animals, even if the area of the brain where the analysis occurs differs [53,54].

## 9. Sustainability

There are many aspects or components of “sustainability”, some components being of part of a wide range of systems while others exist in some systems but not in others. ‘A system or procedure is sustainable if it is acceptable now and if its expected future effects are acceptable, in particular in relation to resource availability, consequences of functioning and morality of action’ [9]. Any one component could make a product unsustainable but a comparison of systems and products should include all components, not just those that allow economic production or impact on carbon dioxide production. In a comparison of beef production systems, the sustainability components for which there were some differences across systems were: human health; welfare of production animals; efficiency of use of world resources; land area per kg meat (with potential consequence for conservation); amount of conserved water per kg meat; greenhouse gas production per kg of meat; extent of water pollution and nitrogen/phosphorus cycle disruption; extent of biodiversity decline; and amount of reduction in carbon sequestration [55]. Although any component with major negative effects can make a system unsustainable, when evaluating systems, the term sustainability should take account of all components. At present, some evaluations consider only one or only a few components so comparisons may not be correct [56,57].

## 10. Conclusions

Some of the meanings of words and definitions of biological concepts that are widely accepted in the scientific community are not properly used in the wider community. Language may lag behind scientific progress to an extent that has a range of important consequences. Some of the biological misunderstandings and misuse of language described here may exacerbate what is perhaps the greatest current problem for the world: excessive concern about humans and too little concern about all other living organisms. In what ways might it be possible to mitigate the most negative effects of academic and public misunderstanding of words and their underlying concepts? Firstly, drawing attention to the problems may encourage discussion and action. Secondly, educational systems may be modified to promote precision and accuracy in the usage of biological terms. Thirdly, scientific writing and reports to governments may be improved.

All people should be scientifically accurate when using words with precise biological meanings, for example, animal, heart, gut, bacteria, welfare, health, pain and sustainability. All people should avoid negativity about the cognitive ability of animals, small animals, and vaccination. The terms instinctive and innate should not be used.

## Data Availability

No new data were created or analysed in this study.

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
