# Peer review of "Language Should Reflect Biological Knowledge"

_animals, 2025, doi:10.3390/ani15172476_

Round 1
Reviewer 1 Report
Comments and Suggestions for Authors
Interesting paper, perhaps worthy of publication, but struggled to see the relevance, with respect to at least two issues: 1) relative novelty and importance of the conclusions found in this particular Manuscript, that haven't already been discussed in other published literature (many of them cited in the References), and 2) how the author ultimately proposes to "enforce", or, at the very least, convincingly persuade such findings to the broader academic community, to the point of successful voluntary implementation of the arguments presented in the Manuscript. Notably, these are not trivial issues, and in my view, deserve at least some attention, presumably in the Discussion and/or Conclusion section of the Manuscript.
Comments on the Quality of English LanguageThe quality of the English language used in the Manuscript, including grammar, spelling and sentence structure, was somewhat below average. Typographical errors were made, verbs were often used at the ends of sentences, and sentences were often structured awkwardly, in ways that were sometimes difficult to understand. For example, line 16 on page 1 of the Manuscript, states: "... in the last twenty tears...", but should presumably state: "... in the last twenty years...". I would therefore highly recommend a very thorough proofreading, editing and review, by a first-language speaker of English and/or professional editor, prior to publication.
Author Response
Reviewer 1
The aims of the paper and the novelty of the paper are now explained better, in the context of other writers, in the extended Introduction to the paper.
The possibilities to bring about change are now briefly outlined in the Conclusions section, now entitled General Discussion and Conclusions
The typographical error has been corrected and the English improved in places.
Reviewer 2 Report
Comments and Suggestions for Authors
Overall Comments
This is an interesting and important discussion on language and biological knowledge. My comments below are meant to reflect more deeply on some of the points made, and the author may elect (or not) to expand some sections.
Line 16: Should read years and not tears
Line 67: In the laws of many countries it is not just animals that are wild or used by humans that are categorised as things, it is all animals.
Also, examples given are animals that are wild, or are used by humans. There is also a differentiation according to whether animals are a pet or a pest, for example pet cats vs wild cats.
Line 109- ‘there is a strong argument for limiting the use of individual names to circumstances where the individual concerned has the capability and opportunity to respond to that name.’
Could you elaborate on why this should be the case? It is not immediately obvious to me why, for example, you might not name a hermit crab you keep as a companion animal if they don’t have the capacity to respond to their name?
Section 3. Genetically Determined, Instinctive, Innate
This section deals with the interaction of genetics and the environment, which in the past was thought of as nature vs nurture. Thinking has moved now to nature AND nurture- that both influence behaviour and the biology of animals. I have given some other references below that could be used, it would be good to acknowledge this discussion in this section.
Line 113: ‘Although all characteristics of plants and animals are affected by genetics, recent evidence from epigenetics research shows that every stage in the expression of genetic material, even the initial production of mRNA, can be affected by environmental factors [15].
Ref #15 refers to post-translational modification of mRNA due to changes in metabolism, and not to epigenetics. Better references for the effect of epigenetics on behaviour would be:
Dion, A., Muñoz, P. T., & Franklin, T. B. (2022). Epigenetic mechanisms impacted by chronic stress across the rodent lifespan. Neurobiology of Stress, 17, 100434.
Jensen, P. (2025). From nature to nurture–How genes and environment interact to shape behaviour. Applied Animal Behaviour Science, 285, 106582.
Line 118: ‘As a consequence, these words should not be used and it is incorrect for a person to excuse their own anti-social behaviour as being in their genes so unavoidable.’
This is called genetic determinism- if the author would like to expand this it would be interesting to talk about genetic essentialism- see Harden, K. P. (2023). Genetic determinism, essentialism and reductionism: semantic clarity for contested science. Nature Reviews Genetics, 24(3), 197-204.
Line 136: It is not clear the point being made in this paragraph and how it is related to the misuse of language and the excessive concern of humans and too little concern about other animals. Is anti-vaccination only a problem in humans, or are there examples in animals? I agree with what is written- it just needs to be linked to the overall premise of the paper more explicitly.
Line 155: ‘Many members of the public are interested in evidence for non-human animals being clever but show reluctance to change language accordingly.’
Is there any evidence to support this statement? We know that human behaviour change is complex, and that choosing to use different words is not a straightforward process. It is also not just the cognitive ability of different species that can vary, but also their perceived emotional capabilities (see Callahan, M. M., Satterfield, T., & Zhao, J. (2021). Into the animal mind: Perceptions of emotive and cognitive traits in animals. Anthrozoös, 34(4), 597-614.)
Line 221: Should read ‘some components being part of..’
Line 222: ‘A system or procedure is sustainable if it is acceptable now and if its expected future effects are acceptable, in particular in relation to resource availability, consequences of functioning and 224 morality of action [7].’
This is a direct quote of the sentence from the chapter cited, and quotation marks are needed.
Author Response
Reviewer 2
Line 16 corrected
Line 67 The sentences at the beginning of the third paragraph have been improved to clarify categorisation as things. The point about problems of specifying according to human usage is also now made.
Line 109 The sentences at the end of section 2 have been modified to explain this point.
Section 3 and Line 113 The wording has been changed and the extra references added.
Line 118 The concepts have been stated and the reference to Hardern added.
Line 136 Several sentences have been added to paragraph 2 of Section 4 to clarify how misunderstanding of words and concepts caused harms.
Line 155 References stating this added. The text has been modified to take account of the information in the references.
Line 221 The papers cited are about components of sustainability but the text has been extended and altered to explain.
Line 222 Quotation marks added.
Reviewer 3 Report
Comments and Suggestions for Authors
The author of the proposed paper adopts--in order for the public to have adequate knowledge of biological phenomena--a very naive perspective, believing that the viewpoint of science is objective and free from bias. The author of the paper does not seem aware that an intervention such as the one he proposes raises serious problems: what authority can decide what is the right way to talk about a certain topic? The author also shows that he has a simplistic idea of how a language works, and how to control it, assuming it is possible. Among other things, since the article deals with language, the author does not seem to have in mind the enormous philosophical and scientific literature on the relationship between thought, language and society.
Author Response
Reviewer 3
The valuable point made by the reviewer has led to several changes in the paper. The introduction to the paper has been extended with extra references and has been improved in several places to consider better the writings of philosophers and others about language and its social impacts. Section 9 of the paper is now General Discussion and Conclusions and is extended in order to explain better and make suggestions about possible action.
Reviewer 4 Report
Comments and Suggestions for Authors
It could be useful to give some examples of countries that have already made some changes in the use of concepts for introducing a balance between humans and other animals. Like presenting the efforts in the law field.
Another aspect to consider, perhaps in the conclusion section, is the proposal of who should make the efforts for this change to happen: is it the scientific community, environmentalists, welfare associations, government agencies? A call should be made to whom?

A central point of the author is misuse of the of language, however, it is recommended to indicate where. The author in some cases gives examples of literature in some science fields (for example, 3. Genetically Determined, Instinctive, Innate) and in others it is not clear if the author is considering the scientific community or the public in general.
Author Response
Reviewer 4
Although some countries have changed laws in relation to animal welfare and some other aspects of sustainability, misuse of biologically defined words is widespread, including in legislation.
The issue of what action might be taken is now mentioned in the expanded General Discussion and Conclusions.
In each area, some comment on misuse is made. It is the view of the author that, while the biggest problems are with the general public, some scientists use biological terms wrongly. Some effort has been made in the paper to give examples but there are many other examples.
Reviewer 5 Report
Comments and Suggestions for Authors
A fantastic and important paper...
- making the world a better place for all fellow creatures
- making very clear, how powerful language can be
- showing, that language affects our behavior towards others and 'the others'!
- showing the many, many fields, where reconsideration is absolutely necessary
- with the high potential of being a 'game changer'
- comprising so many different perspectives
- being an ideal text for all readers and activists interested in non-human animal rights and non-human animal welfare
- using clear language, concepts and statements (without any 'missionary tendency'), which even increases the impact & value
l. 158: some rodents (instead of: rodents = all rodents)
In most of the footnotes the numbers are given 2 times (except of 4, 6, 11, 35). Has this technical issues?
The text is very concise, which is great; but I guess, it could be interesting for the readers to learn a little more about where the old and out-dated words & concepts come from (e.g. Aristotle concerning the sheep; or the ancients concerning the functions of organs): So everybody could see clearly enough, how 'authorities' influenced idioms and as a consequence the view on non-human (= 'the others') and human animals (= 'us').
Please make this text accessible to all sceptics and share it with human-animal-studies-people, with activists, zoologists, vets and physicians: It will help those, who share your opinion, and even broaden und deepen their knowledge; and it will open the eyes, minds, and hearts - sorry for this metaphor - of those, who still believe in the uniqueness of the human species.
Author Response
Reviewer 5
Thanks for the positive comments.
Line 158 The word some has been inserted and the argument clarified.
The history of the use of words is interesting but could make the paper very long and perhaps dilute the message. I have expanded the explanation in several parts of the paper.
References
All new references have been added and references renumbered.
Round 2
Reviewer 1 Report
Comments and Suggestions for Authors
In this revised version of the Manuscript, I note that the Manuscript is now probably worthy of publication, now that my primary reservations, with respect to at least two issues, have now been more-or-less resolved, namely: 1) relative novelty and importance of the conclusions found in the Manuscript, and 2) how the author ultimately proposes to enforce and/or persuade such findings to the lay public and broader academic community. Nonetheless, as described in more detail, below, I would still highly recommend a very thorough proofreading, editing and review, by a first-language speaker of English and/or professional editor, prior to publication.
Comments on the Quality of English LanguageIn this revised version of the Manuscript, the quality of the English language used in the Manuscript, including grammar, diction and sentence structure, was somewhat better. However, verbs were sometimes still used at the ends of sentences, and sentences were often structured awkwardly, in ways that were sometimes difficult to understand. More specifically: 1) many of the sentences were unnecessarily long, and could presumably benefit from added commas and/or breaking up the longest sentences into two sentences, 2) in Section 7, in order to more persuasively make the case for the central argument of the paper, I would recommend that Paragraph 1 be moved to the position of Paragraph 2 (and vice versa), and 3) in Section 10, a few recommendations are presented, but few specific examples are provided; I would, therefore, recommend a few specific examples to more directly flesh out the 3 points presented, here. In other words, the General Discussion and Conclusions section, is conspicuously shorter than the rest of the paper. Lastly, I would also, therefore, recommend a very thorough proofreading, editing and review, by a first-language speaker of English and/or professional editor, prior to publication.
Author Response
Thank you for your comments. If the manuscript is accepted, MDPI’s English editors will review and correct any grammatical errors, and the production team will assist with final proofreading